# CCNE1 Amplification as a Predictive Biomarker of Chemotherapy Resistance in Epithelial Ovarian Cancer

**DOI:** 10.3390/diagnostics10050279

**Published:** 2020-05-05

**Authors:** Justin W. Gorski, Frederick R. Ueland, Jill M. Kolesar

**Affiliations:** 1Division of Gynecologic Oncology, Department of Obstetrics & Gynecology, University of Kentucky Chandler Medical Center, 800 Rose Street, Lexington, KY 40536-0263, USA; fuela0@uky.edu; 2Department of Pharmacy Practice & Science, University of Kentucky College of Pharmacy, 567 TODD Building, 789 South Limestone Street, Lexington, KY 40539-0596, USA; jill.kolesar@uky.edu

**Keywords:** cyclin E1, CCNE1 amplification, predictive biomarker, chemotherapy resistance, ovarian cancer, DNA damage response

## Abstract

Ovarian cancer is the most-deadly gynecologic malignancy, with greater than 14,000 women expected to succumb to the disease this year in the United States alone. In the front-line setting, patients are treated with a platinum and taxane doublet. Although 40–60% of patients achieve complete clinical response to first-line chemotherapy, 25% are inherently platinum-resistant or refractory with a median overall survival of about one year. More than 80% of women afflicted with ovarian cancer will recur. Many attempts have been made to understand the mechanism of platinum and taxane based chemotherapy resistance. However, despite decades of research, few predictive markers of chemotherapy resistance have been identified. Here, we review the current understanding of one of the most common genetic alterations in epithelial ovarian cancer, CCNE1 (cyclin E1) amplification, and its role as a potential predictive marker of cytotoxic chemotherapy resistance. CCNE1 amplification has been identified as a primary oncogenic driver in a subset of high grade serous ovarian cancer that have an unmet clinical need. Understanding the interplay between cyclin E1 amplification and other common ovarian cancer genetic alterations provides the basis for chemotherapeutic resistance in CCNE1 amplified disease. Exploration of the effect of cyclin E1 amplification on the cellular machinery that causes dysregulated proliferation in cancer cells has allowed investigators to explore promising targeted therapies that provide the basis for emerging clinical trials.

## 1. Introduction

Ovarian cancer is the fifth most common cause of cancer death in females in the United States. In the United States in 2020, it is estimated that 21,750 women will be diagnosed with ovarian cancer and that ovarian cancer will be responsible for 13,940 deaths [1]. The fatality rate of ovarian cancer is approximately 70%. Of those deaths, 80% will occur within 5 years of diagnosis [2]. The standard and preferred first-line therapy for stages II–IV ovarian cancer is a carboplatin/paclitaxel doublet in combination with surgical debulking [3,4]. 80% of those afflicted with ovarian cancer will have a recurrence. Diagnosis of recurrence within six months of completion of platinum-based therapy classifies a patient as platinum-resistant. Patients with platinum sensitive recurrence have a median overall survival of 24–36 months. However, median overall survival drops precipitously to 9–12 months for those with platinum-resistant disease. Those who progress on platinum-based chemotherapy are termed platinum-refractory and have a drastically lower median overall survival of 3–5 months [5]. 

Despite a 29% decrease in the overall cancer death rate in the U.S. from 1991–2017, there has not been an appreciable decrease in deaths from ovarian cancer during the last 30 years. This can in large part be attributed to an inability to screen for the disease, overcome resistance to standard-of-care cytotoxic chemotherapy and a failure to identify patients that may benefit from upfront molecularly targeted therapy. Identification of novel molecular subtypes of epithelial ovarian cancer (EOC) from which the response to upfront cytotoxic chemotherapy can be predicted may provide valuable information for the patient and treating physician.

Over the last decade, an emerging subset of EOC tumors driven by cyclin E1 (CCNE1) amplification have been identified as highly prevalent, associated with a poor outcome, possibly predictive of chemotherapy response and conceivably targetable. The primary objective of this study was to review the relationship between CCNE1 amplification and cytotoxic chemotherapy resistance and evaluate CCNE1 amplification as a predictive biomarker in ovarian cancer. Secondary objectives included describing the role of DNA damage response mechanisms in the setting of CCNE1 amplification and a discussion of emerging novel therapeutic agents that may portend a targeted treatment advantage in this subpopulation.

## 2. Role of Cyclin E1 in Normally Dividing Cells

### 2.1. Cyclin E1: Discovery and Importance in the Cell Cycle

Historically, cyclins were first discovered in 1983 as rapidly accumulated and degraded proteins present in dividing sea urchin (*A. punctulata*) embryos [6]. Leland H. Hartwell, R. Timothy Hunt, and Paul M. Nurse won the 2001 Nobel Prize in Physiology or Medicine for the discovery of these “key regulators of the cell cycle” [7]. The term “cyclin” was chosen by Sir R. Timothy Hunt as an homage to his cycling hobby but the name was widely adopted due to the way that the proteins come and go throughout the course of the cell cycle [8]. Over the next decade, these regulatory proteins were isolated, and sub-classified based upon the phase of the cell cycle in which they appear. The 50 kD cyclin E1 protein encoded by the gene CCNE1 was first isolated and described in 1991 in yeast (*S. cerevisiae*) where it was found to bind cyclin-dependent kinase 2 (CDK2) in G_1_-arrested human cell extracts [9]. Shortly thereafter, the cyclic accumulation of cyclin E1 in the G_1_ and S phases was recognized thus positioning cyclin E1 as an important contributor to the G_1_/S cell cycle transition [10].

### 2.2. Role of Cyclin E1 in G1/S Phase Progression

It has since been determined that in order to overcome the G_1_-phase restriction point and enter S-phase, normally dividing eukaryotic cells must specifically activate CDK2 via phosphorylation of a region called the T loop via binding with cyclin E1 [11]. A variety of cellular mechanisms ensure that the cell remains in the G_1_ phase by keeping CDK2 inactive until mitogenic signals intervene. One of these mechanisms is based on limiting the supply of cyclin E [12]. Protein expression levels of Cyclin E are centered around the G_1_/S phase transition and only begin to accumulate in the early G_1_ phase, peak in late G_1_ and again reaches a nadir in G_2_/M [13]. Cyclin E expression is dependent on E2F transcription factors, which are bound to the retinoblastoma protein (Rb) and inactivated when cells are at rest [14,15]. Release of E2F from Rb is accomplished upstream through mitogenic stimuli such as c-Myc, which increase the expression of D-type cyclins that in turn combine with CDK4 and CDK6 to phosphorylate and inactivate Rb along with its complex partners p130 and p107 [16]. Once the cyclin E1/CDK2 complex is activated, it can further upregulate its own expression by phosphorylating Rb and releasing more E2F independent of mitogenic stimuli [17]. This positive feedback loop drives the mitotic transition from the G_1_ to the S phase. 

Once activated, the cyclin E/CDK2 complex is an important regulator of the initiation of DNA replication during the G_1_ phase and centrosome duplication [18]. First, it has been posited that the activated cyclinE1/CKD2 complex is an essential component of the chromatin remodeling process that makes DNA replication competent [19] by stimulation of p300/CBP which relaxes chromatin with its intrinsic histone acetyltransferase (HAT) activity [20]. This epigenetic process remodels chromatin and creates a favorable environment for DNA replication to occur. Second, appropriately regulated levels of cyclin E are also necessary for the formation of the correct number of centrosomes during mitosis. Proper orchestration of these processes is tantamount to ensuring that DNA replication is initiated and that two daughter cells are formed with the correct numbers of chromosomes.

### 2.3. Regulation of Cyclin E1

As S-phase progresses, the dramatic downregulation of cyclin E1 is controlled by CCNE1 promoter silencing and mRNA inhibition. Additionally, intracellular cyclin E1 levels sharply decrease via direct protein degradation [21]. 

The CCNE1 promoter is silenced in two ways. First, the chromobox homolog 7 (CBX7), which is a member of the polycomb repressive complex 1 (PRC1), cooperates with other polycomb group proteins to repress CCNE1 gene transcription directly [22]. CBX7 also epigenetically regulates gene expression via the recruitment of histone deacetylase 2 (HADC2) to the E-box region of the CCNE1 promoter which transcriptionally represses CCNE1 expression [23].

MicroRNA mediated inhibition of cyclin E1 has also been described and may play an important role in the regulation of cyclin E1 levels as the S-phase progresses. MicroRNAs (miRNAs) are 17–22 nucleotide long endogenous noncoding regulatory RNAs that bind to complementary sequences in the 3′-untraslated region (3′-UTR) of the target mRNAs and regulate the expression of target genes at the post-transcriptional level [24]. Specifically, it has been shown that miR-15b [25] and the miR-16 family (miR-16 and miR-424) target CCNE1 mRNA and repress its activity [26]. Interestingly, miR-16 also induces apoptosis by downregulation of the anti-apoptosis protein Bcl-2.

Finally, at the protein level, cyclin E1 is eliminated with the proteasome via ubiquination. The state of cyclin E1 as monomeric or bound to CDK2 determines which ubiquitination pathway is utilized. Monomeric cyclin E1 is ubiquinated by a member of the cullin family of E3 ubiquitin-protein ligases, cul-3 [27]. However, cyclin E1 bound to its CDK2 partner is ubiquinated by the SCF ubiquitin-ligase pathway using the F-box protein substrate binding partner Fbw7 [28]. In addition to ubiquination and degradation, the activity of cyclin E bound to CDK2 is also inhibited directly by binding to the tumor suppressors p21 and p27. 

## 3. Dysregulated Cyclin E1 in Malignancy

### 3.1. Prevalence

CCNE1 amplified tumors account for 19.01% of all ovarian cancer samples included in The Cancer Genome Atlas (TCGA) PanCan 2018 dataset [29]. This makes CCNE1 amplification the eighth most common copy number amplified (CNA) gene present in the ovarian tumors analyzed. CCNE1 amplification is also frequently present in a variety of other histologic subtypes and sites of primary disease. CCNE1 amplification is especially prevalent in gynecologic malignancies and composes 40.35% of uterine carcinosarcoma samples and 7.56% of other uterine histologic subtypes included in the TCGA PanCan 2018 datasets [30,31] (Figure 1).

### 3.2. Mechanisms of CCNE1 Amplification Driven Oncogenesis

Overexpression of cyclin E1 increases the speed that cells progress through the G_1_/S-phase restriction point which leads to genomic instability. Constitutive overexpression of cyclin E in vitro has been shown to increase the frequency of polyploid cells [32,33]. These abnormalities in chromosomal segregation during cell division increases the rate of chromosomal mutations in other genes that control cell survival and proliferation. In turn, affected cells push toward a tumorigenic phenotype. 

Overexpression of CCNE1 adversely affects DNA replication by disrupting origin firing and replication progression [34]. This results in oncogene-induced replication stress [35]. In the setting of cyclin E overexpression, the frequency of polyploid cells produced during mitosis dramatically increases [32]. Mutational upregulation of cyclin E leading to high activity of CDK2 has been shown to induce centrosome amplification concomitantly with p53 loss [36,37]. In turn, the cells produced are polyploid with multiple centers of partially completed mitosis. Recently, a low molecular weight variant of cyclin E has been found to have an even more profound effect on this centrosome-mediated deleterious process [38]. Collectively, these abnormalities are omnipresent in cyclin E1 amplified tumorigenesis and effectively increase the rate of chromosome mutations in genes that control cellular proliferation and survival (Figure 2).

## 4. Chemotherapy Resistance

CCNE1 amplification has been identified as a primary oncogenic driver in a subset of high grade serous ovarian cancer that have an unmet clinical need due to resistance to standard of care and targeted chemotherapeutic agents. Consideration of the interplay between cyclin E1 amplification and other common ovarian cancer genetic alterations provides the basis for understanding chemotherapeutic resistance in CCNE1 amplified disease. In fact, the rationale for using CCNE1 as a predictive biomarker of chemotherapy resistance stems from insight into the mechanism of CCNE1 amplified oncogenesis and its relationship to DNA damage response pathways and mitotic progression. 

### 4.1. Poly (Adenosine Diphosphate (ADP)-Ribose) Polymerase (PARP) Inhibitors

Interestingly, homologous recombination pathway mutations and CCNE1 amplification are nearly mutually exclusive. Homologous recombination pathway deficient (HRD) tumors are associated with germline and somatic variants in BRCA1, BRCA2, RAD51, PALB2 as well as approximately ten other “BRCAness genes” [40] and comprise approximately 50% of high grade serous ovarian cancers (HGSOC) [29]. HRD status provides the rationale for the use of PARP inhibitors and is predictive of platinum-based chemotherapy sensitivity.

The CCNE1 amplified subgroup is unlikely to benefit from advances in the treatment of HGSOC with DNA repair pathway targeted agents such as PARP inhibitors. Mechanistically, PARP inhibitors target and inhibit the single stranded DNA break (SSB) repair pathways: base excision repair (BER), nucleotide excision repair (NER) and mismatch repair (MMR). Thus, when cancer cells are exposed to PARP inhibitors, single stranded DNA breaks turn into double stranded breaks [41]. In HRD tumors, these double stranded breaks are not able to be repaired by the high-fidelity homologous recombination (HR) pathway and instead are repaired with the error prone non-homologous end joining (NHEJ) pathway. Cells dependent on the NHEJ pathway are not able to replace missing nucleotides but instead join strands of DNA to repair the break. This results in an inexact replica of the normal DNA sequence, frequent errors and an accumulation of mutations that eventually lead to genomic instability and cell death [42]. Since HRD and CCNE1 amplification rarely co-occur, it is usually not possible to exploit alterations in the DNA repair pathways to produce synthetic lethality using PARP inhibitors in the cyclin E1 amplified subgroup (Figure 3). 

### 4.2. Platinum Agents

The response to platinum-based agents, such as carboplatin and cisplatin, is also more likely to be muted in the CCNE1 amplified but HR proficient subset. Platinum agents cause a variety of DNA lesions by way of creating intrastrand and interstrand crosslinks [43]. These adducts block transcription and DNA synthesis, which can result in cell cycle arrest and cell death. Most of the major DNA repair systems are involved in removing platinum-induced DNA damage including the NER, MMR, HR and NHEJ pathways [44]. Activation of these intact pathways decreases the efficacy of platinum-based agents and contributes to platinum resistance. The mutual exclusivity of HRD and CCNE1 amplification all but ensures proficiency of the highly accurate HR pathway in CCNE1 amplified tumors. Thus, CCNE1 tumors are more likely to be resistant to standard of care platinum-based cytotoxic agents (Figure 3).

### 4.3. Taxanes

The antimitotic agent paclitaxel combined with carboplatin forms the backbone of the frontline treatment of ovarian cancer. Taxanes bind β-tubulin, which in turn stabilizes microtubules and prevents the normal formation of mitotic spindles [45]. This alteration in microtubule dynamics produces a prolonged mitotic arrest via activation of the spindle assembly checkpoint. This extended mitotic arrest causes tumor cells to either undergo death in mitosis or exit mitosis in a tetraploid G1 state, which is described as mitotic “slippage” [46]. Although ovarian cancers are usually responsive to paclitaxel, acquired drug resistance via increased mitotic slippage is common [47]. CCNE1 amplification has been linked with higher levels of low molecular weight cyclin E (LMW-E), which can cause faster mitotic exit and an increased rate of mitotic slippage [48]. 

## 5. CCNE1 Amplification as a Biomarker

### 5.1. Prognostic Biomarker

Several studies have correlated CCNE1 amplification in epithelial ovarian cancer with poor outcomes. One such study identified CCNE1 amplification in 18 (20.4%) of 88 ovarian carcinomas included in the study and found that the presence of CCNE1 amplification correlated with shorter disease-free survival and overall survival (*p* < 0.001). Multivariate analysis demonstrated that CCNE1 gene amplification status was an independent prognostic factor for disease-free survival and overall survival after standard platinum-taxane chemotherapy (*p* = 0.0274, *p* = 0.0023) [49]. 

Other primary disease site agnostic studies have also linked CCNE1 amplification with worse outcomes. A recent study attempted to elucidate the biological basis of worse cancer survival outcomes in the African American population. The genetic ancestry of subjects included in The Cancer Genome Atlas were estimated and a pan-cancer analysis of the influence of genetic ancestry on genomic alterations was performed. African Americans with breast, head and neck, and endometrial cancers were found to exhibit a higher level of chromosomal instability compared with European Americans. The frequencies of TP53 mutations and amplification of CCNE1 were increased in African Americans in the tumors with higher levels of chromosomal instability [50]. Thus, CCNE1 amplification seems to be a valuable prognostic biomarker in ovarian cancer and across other cancer subtypes. 

### 5.2. Predictive Biomarker

#### 5.2.1. Ovarian Cancer

Few studies have specifically investigated CCNE1 amplification status as a predictive biomarker of chemotherapy response in ovarian cancer. One study found that CCNE1 amplification correlates with chemoresistance in ovarian cancer. The study measured genome-wide copy number variation in 118 ovarian tumors using high-resolution oligonucleotide microarrays. The copy number variation was then compared to a well-defined subset of 85 advanced-stage serous tumors to deduce primary resistance to treatment. It was determined that amplification of the 19q12 region of the genome, which contains CCNE1, was significantly associated with a poor response to primary treatment [51].

Conversely, while not exactly a study of CCNE1 amplification per se, another study found that cyclin E1 overexpression did not correlate with pathologic response to neoadjuvant chemotherapy [52]. Saponzik et al. [52] examined the role of cyclin E1 positive-immunostain as a predictor of first-line taxane-platinum chemoresistance. Matched pre- and post-neoadjuvant chemotherapy tumor samples with and without cyclin E1 overexpression were correlated with the degree of pathological response to treatment using chemo-response scores. In this subset of patients, it was found that cyclin E1 immunohistochemistry did not predict taxane-platinum chemoresistance in ovarian cancer patients. It should be noted that the well-accepted notion that gene amplification contributes to increased expression still remains a reasonable but unproven assumption [53]. Thus, CCNE1 amplification, but not cyclin E1 overexpression, is a more reliable predictive biomarker of chemotherapy resistance in epithelial ovarian cancer.

#### 5.2.2. Other Primary Disease Sites

Despite the scarcity of studies investigating CCNE1 amplification as a predictive biomarker in ovarian cancer, other primary disease sites have linked cyclin E1 amplification with poor response to chemotherapy both in vitro and in vivo. 

Breast: Secondary analysis from the PALOMA-3 trial [54] demonstrated that the efficacy of the CDK4 and CDK6 inhibitor palbociclib was lower in patients with high cyclin E1 (*CCNE1*) mRNA expression. The median progression free survival (PFS) in the palbociclib arm was 7.6 months in the high CCNE1 mRNA expression group versus 14.1 months in the low CCNE1 mRNA expression group. Interestingly, high levels of CCNE1 mRNA expression was more predictive of resistance to palbociclib in metastatic tumors than in archival primary biopsy tissue samples. Thus, high CCNE1 mRNA expression levels may be an effective predictive biomarker of resistance to palbociclib in hormone-receptor-positive, HER2-negative metastatic breast cancer that has progressed on previous endocrine therapy.Multiple Myeloma: Incubation of various multiple myeloma cell lines with seliciclib, a selective CDK2/E, CDK2/A, CDK7 and CDK9-inhibitor, resulted in apoptosis. However, ectopic over expression of CCNE1 resulted in reduced sensitivity of the multiple myeloma tumor cells in comparison to the paternal cell lines. Conversely, silencing of CCNE1 with siRNA increased the cell lines’ sensitivity to seliciclib [55].Bladder: A cisplatin sensitive human bladder cancer cell line (T24) and a cisplatin resistant bladder cancer cell line (T24R2) were analyzed using microarray to determine the differential expression of genes that are significant in cisplatin resistance [56]. CCNE1 was found to be one of 18 significantly upregulated genes detected in vitro. Western blot analysis confirmed higher levels of the protein products of three of the 18 upregulated genes including CCNE1. Thus, CCNE1 upregulation seem to be a predictive biomarker of cisplatin resistance in bladder cancer.

## 6. Emerging Targeted Therapies in Clinical Trials

Until this year, there were no clinical trials in which patients were selected based on *CCNE1*-amplification status. This may be primarily due to the fact that there are no drugs currently available that specifically target Cyclin E1 amplification. In fact, the development of small molecule inhibitors that directly target cyclins is unlikely and potentially not feasible. Cyclins act as regulatory subunits and thus are not as easily directly targeted as an enzyme or receptor [57]. However, several alternative approaches have been proposed to combat *CCNE1*-amplified tumors.

### 6.1. Cyclin Dependent Kinase (CDK) Inhibitors

CDK2 is an attractive target in treating CCNE1 amplified tumors due to its relative specificity for cyclin E and the essential role CDK2 plays in the activated CDK2/cyclin E1 complex. The targeted inhibition of CDKs with pan-CDK inhibitors, and more specific CDK2 inhibitors, have been extensively explored and show promise in CCNE1-amplified malignancies in vitro [58,59]. In fact, several pan-CDK inhibitors with CDK2-specific activity have been evaluated in clinical trials including: AT7519 (AT7519M, Astex Therapeutics Ltd., Cambridge, UK), AG-024322 (Pfizer), Dinaciclib (MK7965, SCH727965, Merck & Co), CYC065 (Cyclacel Pharmaceuticals), Ronaciclib (BAY 1000394, Bayer), TG02 (Tragara Pharmaceuticals), Milciclib (PHA 848125, Tiziana Life Sciences). Despite the theoretical promise of CDK inhibitors in cyclin E1 amplified disease, none of these agents have progressed beyond Phase II clinical trials. Some trials have been withdrawn due to the associated side effects of off-target kinase inhibition and failure to reach an appropriate clinical outcome [60]. Furthermore, CDK inhibitors have not been studied clinically in ovarian cancer. Advancement of these promising agents may rely on combination strategies, which may both increase efficacy and allow lower dosing to mitigate side effects. The combination of CDK2 and PI3K inhibitors have shown efficacy in glioma and colorectal cancer xenografts [61]. Additionally, CDK2 inhibition has been found to be synergistic with the non-taxane anti-mitotic agent erilbulin in triple negative breast cancer in vitro and in xenograft models. Thus, CDK2 inhibitors may restore chemosensitivity in chemoresistant triple negative breast cancer cases [62].

### 6.2. Wee1 Inhibitors

Specific targeting of the Wee1 kinase may also hold promise as a therapeutic option when cyclin E1 is amplified. When cells are exposed to DNA damaging agents, WEE1 kinase activation occurs and acts to inhibit CDK1, which prevents cell division at the G_2_/M phase checkpoint until the damage is repaired. Activated WEE1 kinase halts premature mitosis and prevents extensive DNA damage and cell death [63]. Targeted inhibition of WEE1 kinase, in conjunction with cyclin E1 amplification, may cause rapid, unregulated mitosis via dysregulation of both the G_1_/S and G_2_/M phase checkpoints. In turn, massive DNA damage will accumulate in CCNE1 amplified cells and genomic instability will lead to preferential cell death in tumor cells. Due to the theoretical promise of targeting these interrelated pathways, a phase II clinical trial of the WEE1 kinase inhibitor, AZD1775, in CCNE1 amplified disease is now underway. To our knowledge, this is the first biomarker driven clinical trial that explicitly includes CCNE1 amplification. 

## 7. Discussion

Cyclin E1 amplification is a common and distinct driver of tumorigenesis in epithelial ovarian cancer (EOC) that is associated with poor response to standard of care upfront chemotherapeutic agents. We propose three innovative strategies to meet the clinical needs of EOC patients with CCNE1 amplification.

### 7.1. Increase Somatic Molecular Profiling with a Broad Panel of Actionable Mutations Using Specimens Obtained at the Time of Initial Diagnosis or Primary Debulking Surgery

At this time, stratification of epithelial ovarian cancer patients into molecularly classified cohorts still lags behind treatment decisions. Thus, CCNE1 amplified disease is likely to be significantly underappreciated. In fact, the National Cancer Care Network (NCCN) guidelines state that molecular tumor testing is recommended prior to initiation of therapy for persistent/recurrent disease using the most recent available tumor tissue including at least: BRCA1/2, homologous recombination pathway genes, and microsatellite instability or DNA mismatch repair [4]. Unfortunately, use of a narrow panel that only satisfies NCCN recommendations would fail to identify CCNE1 amplified disease. Furthermore, without explicit recommendations for molecular profiling in the upfront setting, the window for the maximum benefit of targeted therapy in CCNE1 amplified disease may be missed. The underlying genomic drivers of disease response, such as CCNE1 amplification, may not be appreciated until the patient is well into therapy, if at all.

### 7.2. Encourage Enrollment of ccne1 Amplified Patients into Disease Site Agnostic Biomarker Driven Basket Trials

Basket trials, such as the one designed for CCNE1 amplified disease using the WEE1 kinase inhibitor AZD1775, hold promise to improve outcomes. Two other classes of agents have recently received FDA approval for all solid tumors with specific genetic alterations regardless of primary disease site. As of May 2017, pembrolizumab is approved for use with microsatellite instability-high (MSI-H) or mismatch repair deficient (dMMR) solid tumors regardless of histology or primary disease site. This landmark recommendation was the first time that the FDA had approved a cancer treatment for an indication based on a common biomarker rather than the primary site of origin [64]. Following suit, in November 2018, the tropomyosin receptor kinase (TRK) inhibitors Vitrakvi (larotrectinib sulfate) and Rozlytrek (Entrectinib) received approval for use in solid tumors with a neurotrophic receptor tyrosine kinase (NTRK) gene fusion [65]. Unfortunately for patients afflicted with CCNE1 amplified disease, neither of these agents target interrelated pathways that were driven by high levels of cyclin E1. However, approval of these agents represents a paradigm shift in the treatment of cancer and holds promise for FDA approval of more primary site agnostic biomarker-driven targeted drugs.

### 7.3. Continue In Vitro, In Xeno and In Vivo Investigation into Combinatorial Treatment Strategies That Exploit Interrelated Pathways to Directly Inhibit Tumor Growth or Re-Sensitize CCNE1 Amplified Disease to Cytotoxic Chemotherapy

Finally, it has yet to be determined if WEE1 kinase inhibitors or CDK inhibitors alone will show efficacy against CCNE1 amplified disease. However, if these agents in isolation are not able to achieve improved outcomes then the logical next step in clinical trial design would be to combine these agents with each other and standard cytotoxic chemotherapy. Combining a WEE1 inhibitor or a CDK2 inhibitor with dose-dense paclitaxel in the recurrent setting may be a good starting point to combat standard of care cytotoxic chemotherapy resistance. Another more experimental approach may require reliance on insights gleamed from basic science in vitro and in vivo work. For example, targeting Cyclin E1 by a PLK1 inhibition-based stabilization of FBW7 might be a novel approach to increase apoptosis in *CCNE1*-amplified ovarian cancer cells [66,67]. Another strategy may involve the use of active ataxia telangiectasia mutated and rad3-related (ATR) kinase inhibitors in CCNE1 amplified and TP53 mutated cancers. CCNE1 amplification exaggerates the hypersensitivity of TP53 deficient cells to ATR inhibition [68].

## 8. Conclusions

Exploring the interplay between cyclin E1 amplification and other common ovarian cancer genetic alterations provides the basis for understanding chemotherapeutic resistance in CCNE1 amplified disease. The near mutual exclusivity of homologous recombination pathway mutations and CCNE1 amplification results in resistance to platinum-based cytotoxic chemotherapies and ineffective PARP inhibition. Higher levels of low molecular weight cyclin E (LMW-E) in CCNE1 amplified tumors can cause faster mitotic exit, an increased rate of mitotic slippage and resistance to anti-mitotic chemotherapies such as taxanes. 

CCNE1 amplification has been identified as a predictive biomarker of poor chemotherapy response in epithelial ovarian cancer and other cancers. However, cyclin E1 protein overexpression may not be a reliable predictive biomarker of chemotherapy resistance. Thus, somatic sequencing should be carried out to identify the CCNE1 amplified subset of patients. 

Promising targeted strategies using WEE1 kinase inhibitors and CDK2 inhibitors are currently being examined in ongoing biomarker driven clinical trials. The results of these trials may help gynecologic oncologists improve outcomes in this very aggressive and chemoresistant CCNE amplified ovarian cancer cohort that currently has an unmet clinical need.

## Figures and Tables

**Figure 1 diagnostics-10-00279-f001:**
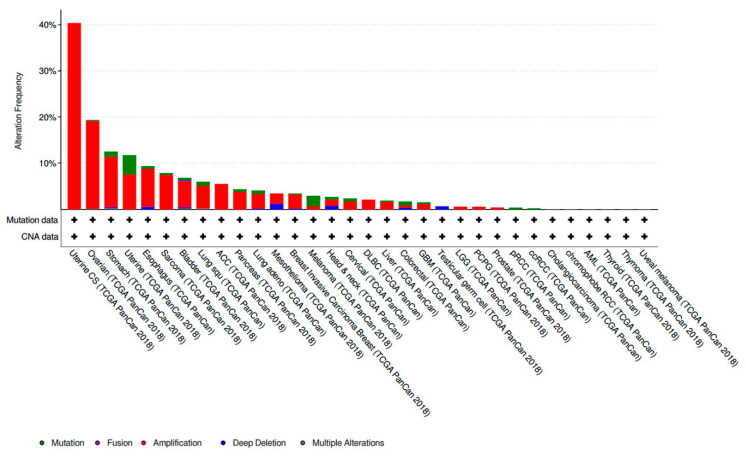
Prevalence of CCNE1 amplification in The Cancer Genome Atlas (TCGA) PanCan 2018 data sets across a variety of primary disease sites and histologic subtypes [30,31].

**Figure 2 diagnostics-10-00279-f002:**
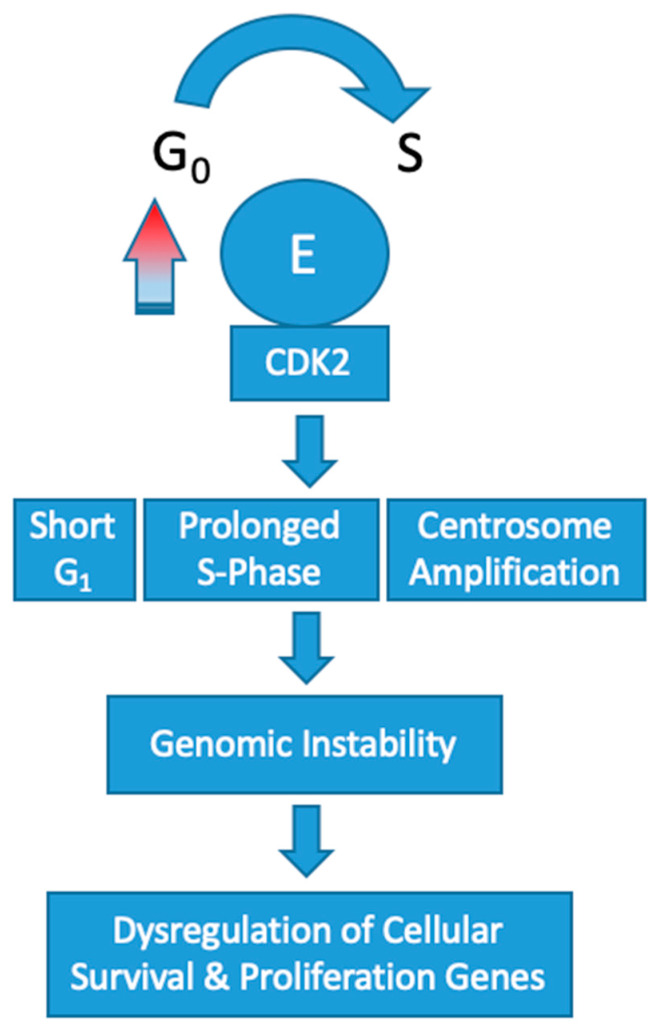
Schematic representation of the impact of CCNE1 amplification on oncogenesis. Adapted from Hwang, H. et al. [39].

**Figure 3 diagnostics-10-00279-f003:**
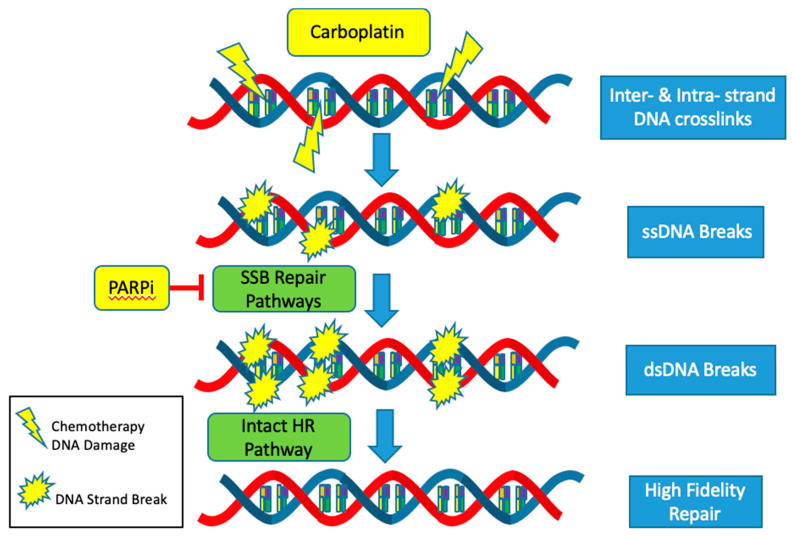
Schematic representation of the mechanism of CCNE1 amplification resistance to front-line cytotoxic (Carboplatin) and targeted (PARP Inhibitors) ovarian cancer chemotherapies.

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
