# Peer review of "CCNE1 Amplification as a Predictive Biomarker of Chemotherapy Resistance in Epithelial Ovarian Cancer"

_diagnostics, 2020, doi:10.3390/diagnostics10050279_

Round 1

Reviewer 1 Report

The authors have reviewed the literature and attempted to handle the topic of the role of CCNE1 amplification as a predictive marker of chemotherapy resistance in epithelial ovarian cancer. As indeed ovarian cancer is the most aggressive gynecologic type of tumor with a very high fatality rate works presenting or reviewing knowledge on this topic are interesting.

The author's work is interesting, is well-written and well-organized. Also, they have made a good attempt in presenting their work. Their figures are interesting and cover the topic well.

I only have some minor comments to be addressed.

Please clarify and elaborate a little bit more on the relation between CCNE1 amplification and CCNE1 over-expression. Are these two related? is amplification an etiological mechanism for CCNE1 over-expression?

Thus, the authors should add a paragraph in the conclusions section on the potential applications of a CCNE1-related inhibitor for endothelial ovarian cancer. How is an inhibitor expected to function based on the over-expression status of CCNE1?

The authors sholuld also highlight their review findings either in the discussion section or the conclusions section.

Reviewer 2 Report

In this manuscript, Gorski and colleagues highlight the relevance of cyclin E in epithelial ovarian cancer. They describe how the cyclin E amplification can influence progression of this cancer and therapies efficiency and propose this protein amplification as biomarker of ovarian cancer. The review is well written, sometimes heavy to read but, overall clear and abundant in information. Maybe, the discussion could be written in a more incisive way, for more drawing attention to the pivotal role of cyclin E in all the three examined points.

Minor revisions.

Lane 21: "ovarian cancer" is repeated twice.

Lanes 100-101: the authors describe how the cyclin E down-regulation occurs. Please modify the sentence to clarify and differentiate on one side the reduced synthesis and on the other the increased degradation.

Lane 127: amplification and not amplificiation.

In fig 2, the authors should specify that the genomic instability involves genes controlling cellular proliferation and survival (as they assert in lanes 148-149).
